# The Impact of Caregiver Pressure to Eat on Food Neophobia in Children with Autism Spectrum Disorder: A Cross-Sectional Study

**DOI:** 10.3390/children11050528

**Published:** 2024-04-28

**Authors:** Qingqing Xie, Cuiting Yong, Caihong Xiang, Yue Xi, Jiaqi Huo, Jiajing Liang, Hanshuang Zou, Yunfeng Pan, Minchan Wu, Qian Lin

**Affiliations:** Department of Nutrition Science and Food Hygiene, Xiangya School of Public Health, Central South University, 172 Tongzipo Road, Changsha 410031, China; xieqingqing@csu.edu.cn (Q.X.); yongcuiting@csu.edu.cn (C.Y.); xch0622@csu.edu.cn (C.X.); xiyue0404@csu.edu.cn (Y.X.); huojiaqi0611@csu.edu.cn (J.H.); ljj1996@csu.edu.cn (J.L.); zouhanshuang@csu.edu.cn (H.Z.); panyunfeng@csu.edu.cn (Y.P.); wuminchan@csu.edu.cn (M.W.)

**Keywords:** autism spectrum disorders, food neophobia, feeding behavior

## Abstract

(1) Background: With autistic children’s high pervasiveness of eating problems and inappropriate feeding behaviors by their caregivers, this study wanted to inspect the connection between caregivers’ pressure to eat and food neophobia in these children. (2) Methods: Cross-sectional overview of 160 guardians of kids aged 2 to 7 years. After one-on-one questioning by the researcher, the collected information on the socio-demographic characteristics of the children with autism, caregiver feeding behavior, and new food neophobia (FN) scores was entered into the Questionnaire Star system. (3) Results: The mean FN score was 25.56 ± 6.46. The caregiver’s pressure to eat positively related to children’s FN (β = 0.164 95% CI, 0.078, 2.163). In these children, we found a negative correlation between FN score and the frequency of vegetable intake (*p* ≤ 0.001), fruit intake (*p* ≤ 0.05), aquatic product intake (*p* ≤ 0.05), and dietary diversity score (*p* ≤ 0.01), and positively correlated with the frequency of snack intake (*p* ≤ 0.05). (4) Conclusions: Caregiver pressure to eat was positively associated with high levels of FN in Chinese kids with ASD, which in turn negatively impacted dietary quality. To improve eating habits, caregivers should reconsider their feeding strategies and avoid using forceful methods to ease food neophobia in these children.

## 1. Introduction

Children with autism spectrum disorder (ASD) are more susceptible to eating behavior problems such as food neophobia, picky eating and favoritism due to sensory sensitivities, repetitive stereotypic behaviors and narrow interests and activities. A study of 3072 children found that 70.4% had eating behavior problems compared to typically developing (TD) children [1]. Food neophobia, which means “fear of new foods”, was proposed in 1992 by Canadian scholars Prof. Pliner and Prof. Hobden [2]. A total of 58–67 percent of these children have food neophobia [3,4]. They dislike fresh fruits, vegetables, and protein-rich foods in favor of sweets, high-fat and high-carbohydrate foods, and processed foods [5,6,7]. This severely limits the diversity of their food intake and the quality of their diets, resulting in deficiencies in nutrient intake [8]. In a study of Arab children, Al-Farsi et al. found that serum folate and vitamin B12 levels were significantly lower in autistic children than in TD children. They correlated with dietary intake [9]. In addition to vitamins, children with ASD have deficiencies in mineral intake [10,11]. Many nutrients assume a significant part in the advancement of neural networks, and nutrient deficiencies in children with ASD may lead to a vicious cycle of worsening clinical symptoms, which in turn may further affect their eating behavior.

Numerous research has shown a connection between children’s food neophobia and parentally controlled feeding behavior [12,13,14,15]. Therefore, we hypothesized a similar association in children with ASD. For autistic children, caregivers may be overly concerned about their nutritional health and inadvertently resort to pressure to eat, which can also exacerbate children’s fear of new foods. According to studies [16,17], over 60% of parents expressed worries about their child’s development and nutrition. Children with ASD also tend to eat under stress and are more likely to use inappropriate feeding techniques [18,19], such as authoritarian, permissive, and emotional feeding [20,21,22]. On the other hand, shared dishes have always been the core of the family. It has long been part of Chinese food culture and family life. Chinese parents usually do not prepare individual meals for each child but make dishes shared by the whole family [23,24]. However, children’s food preferences may differ from those of adults. Children with autism may exhibit more extraordinary refusal to share dishes due to their stereotyped food choices. There can also be more stress for caregivers trying to get them to eat properly. The characteristics derived from Chinese food culture are also a great challenge for children with ASD and their caregivers.

It has been proven that it is easier to change the feeding practices of parents than to develop interventions to improve the food neophobia of the children directly [25,26]. The sensory sensitivities and unique challenges that individuals with ASD encounter in their relationship with food make them particularly susceptible to the potential harm associated with pressure to eat, making it crucial to investigate and address this issue within this specific population. Understanding these dynamics can help inform more effective strategies to support healthy eating behaviors in these special children. Therefore, this study centers around the relationship between the feeding behaviors of caregivers of autistic children and their food phobia. It also fills in the research on this topic in China. The main research aims are as follows: (a) assess these children’s food neophobia level; (b) examine the connection between FN in children and the pressure from caregivers to eat; and (c) explore the link between FN and dietary intake.

## 2. Materials and Methods

### 2.1. Sample and Procedures

This was a cross-sectional study. The survey involved six special education institutions from different urban areas of Changsha, involving 160 ASD children and their primary caregivers. After obtaining informed consent from the primary caregivers, the investigator completed the questionnaire through one-on-one questioning. The following inclusion and exclusion criteria were used to include subjects in the study: (1) Inclusion Criteria: ① ASD was diagnosed by a specialist in a tertiary hospital; ② Between 2 and 7 years of age; ③ Parents who completed the questionnaire are the primary caregivers of the child (responsible for the child’s diet and care for the child for six months or more); ④ The child’s primary caregiver has regular expression and comprehension skills, gives informed consent and participates voluntarily. (2) Exclusion criteria: The child suffers from ① other developmental disorders or mental illnesses; ② Serious congenital diseases; ③ Acute or chronic infectious diseases in the last three months; ④ Having a high fever, diarrhea, or other ailments that interfere with eating during the survey period. The present study is based on a research project approved by the Medical Ethics Committee of the Hunan Provincial Institute of Maternity and Childhood Healthcare on 29 May 2019. The research project is titled “Metabolomics and Nutritional Management of Autism Spectrum Disorder”. It is a sub-theme of the Hunan Provincial Major Project on Collaborative Prevention and Treatment of Birth Defects in Science and Technology.

### 2.2. Measures

#### 2.2.1. Demographic Variables

Questionnaires were administered to collect basic demographic information on children with ASD, including information on the child’s gender, age, ethnicity, place of residence, whether the child was an only child, birth status, and exclusive breastfeeding duration; basic demographic information on the primary caregiver, including information on gender, age, occupation, and level of education.

#### 2.2.2. Food Neophobia in ASD Children

We used the Child Food Neophobia Scale (CFNS) to assess children’s fear of new food [27]. In 2018, Zou J introduced CFNS to China. The Cronbach’s α was 0.910 [28]. Six items on a 7-point scale, with response options ranging from “strongly disagree to strongly agree” (indicating a score range of 1–7), make up the Chinese version of the CFNS—the first and sixth entries inverted scoring. The overall score for these six elements is known as the CFNS total, and it ranges from 6 to 42, with higher values denoting higher levels of FN.

#### 2.2.3. Caregiver’s Feeding Behavior

The method of evaluating caregiver feeding behavior consists of two parts. The feeding behavior of caregivers was evaluated using the Child Feeding Questionnaire (CFQ) developed by Birch in 1994 [29]. The survey was conducted using the Chinese version of this questionnaire, which was created by Professor Zhu Daqiao’s team in 2016. The research team extracted two entries from the Restricted Diet dimension to form the Food as Rewards (FR) dimension. The Cronbach’s alpha coefficients for the four dimensions (pressure to eat, monitoring, restriction, and food as a reward) of the Chinese version of the CFQ range from 0.695 to 0.867 [30]. The score for each dimension is the mean score of the entry to which it belongs, with higher scores indicating more parental control. Also, forced feeding behavior was captured through self-administered entries about whether parents consistently offered their children the refused foods.

#### 2.2.4. Food Intake in ASD Children

The frequency of food intake in the past three months by children with ASD was investigated by asking the primary caregiver of the child with ASD to complete the Food Frequency Questionnaire (FFQ). There are 27 food items in the FFQ, including cereals, fruits, vegetables, legumes and products, eggs, milk and dairy products, nuts, meat, aquatic products, fats and oils, and snacks. The response formats were “never”, “<1 time/month”, “1–3 times/month”, “1–2 times/week”, “3–4 times/week”, “5–6 times/week”, “1 time/day”, “2 times/day”, and “3 or more times/day”. The dietary frequency of each food group was recorded as the number of times consumed per week: monthly frequency less than 1 coded as 0 per week, 1 to 3 times a month coded as 0.5 times a week, 1 to 2 times a week coded as 1.5 times a week, 3 to 4 times a week coded as 3.5 times a week, 5 to 6 times a week coded as 5.5 times a week, 1 time a day coded as 7 times a week, 2 times a day coded as 14 times a week and 3 times a day coded as 21 times a week. We calculated dietary diversity scores (DDS) based on the type of food consumed by each subject in an average week, with 1 point for each type of food consumed, 2 points for every two types of food consumed, and so on, and sugary beverages, fast food, and snacks were excluded from the scoring. Higher scores indicate higher levels of dietary diversity.

#### 2.2.5. Social Interaction and Behavioral Skills in ASD Children

The Social Responsiveness Scale™, Second Edition (SRS™-2)—Preschool Form for Investigating Social Deficits in Children with ASD was used [31]. The scale consists of 65 items that can be rated by either the primary caregiver or the teacher, and the primary caregiver-reported outcomes were used in this study. All entries were scored on a four-point Likert scale with the following responses: does not meet, somewhat meets, often meets, and almost always meets, with scores from 0 to 3, respectively. The higher score represents the lower social skills.

### 2.3. Data Analyses

We used SPSS 25.0 statistical software (I.B.M. Corp., Armonk, NY, USA) and GraphPad Prism 8 software (La Jolla, CA, USA) for statistical descriptions, Pearson correlation analyses, and multiple linear regression analyses. Pearson’s correlation analysis was used to examine relationships between food neophobia and caregiver feeding practices in children with ASD and dietary intake. Lastly, we investigated the connection between FN and caregiver pressure to eat behavior in children with ASD using multiple linear regression analyses. *p* < 0.05 was deemed statistically significant in this investigation.

## 3. Results

### 3.1. General Characteristics of Children with ASD

Table 1 describes the characteristics of the samples. We investigated 160 children, and their average age was 4.23 ± 1.36 years old. Of these children, 78.8% were boys, 88.1% lived in urban areas, 48.7% were only children, and 44.6% were exclusively breastfed for over six months. 43.1% of caregivers had a bachelor’s degree or higher. Children had the highest level of food neophobia when the primary caregiver’s literacy level was high school. In children with ASD, correlations were discovered between behavioral competence, social interaction scores, and the degree of food phobia.

### 3.2. General Characteristics of Primary Caregiver

In this study, 96.7% of the primary caregivers of children with ASD were female. 56.9% of the primary caregivers were worried about their children being malnourished, and parents who were worried about their children being malnourished had higher scores for forced feeding (Table 2).

### 3.3. Correlation between Food Neophobia and Caregiver’s Feeding Behavior in Autistic Children

The mean score of the children’s FN was 25.56 ± 6.46. We used Pearson correlations to analyze the relationship between FN scores and caregiver feeding behavior (Table 3). The children’s FN scores with autism were correlated with eating pressure (*p* < 0.05) and continuing to provide them with foods they are not allowed (*p* < 0.01). No association was found between the other three feeding behaviors and food neophobia.

### 3.4. Correlation between Food Neophobia and Caregiver’s Feeding Behavior in Autistic Children

Multiple linear regression was used to assess the link between FN and eating pressure in children diagnosed with ASD (Table 4). Five variables, Restriction, Monitoring, Pressure to eat, Food as rewards and Continuing to provide children refuse food, were included in the initial regression model of this study, and stepwise regression was used for variable selection. When adjusted for relevant covariates, the ASD children’s FN score was related to the pressure to eat (β = 0.164, *p* < 0.05) and the continued provision of denied foods to children (β = 0.211, *p* < 0.01).

### 3.5. Correlation between Food Neophobia and Dietary Intake in Autistic Children

The children’s meal frequency scores and FN scores were correlated using Pearson correlation (Figure 1). The frequency of aquatic products (*p* < 0.05), fruits (*p* < 0.05), vegetables (*p* < 0.001), and dietary diversification (*p* < 0.01) is inversely correlated with the score. Additionally, there is a favorable correlation (*p* < 0.05) between it with the frequency of snacking. 

## 4. Discussion

Our objective was to investigate the association between autistic children’s FN and the dietary intake and pressure-to-eat practices of caregivers. We also explored whether caregivers’ pressure-to-eat practices might influence children’s food preferences by affecting their food neophobia and, consequently, their food choices. The degree of food phobia in children with ASD was positively correlated with the pressure from caregivers to eat and with the severity of social impairments. Caregivers who are more worried about their children’s nutritional status are more likely to engage in pressure-to-eat behaviors and are more likely to result in high levels of FN in children. Additionally, we observed a positive correlation between these kids’ FN levels and snack intake and a negative correlation with their intake of fruit, vegetables, and aquatic products.

Additionally, we discovered that dietary diversity and FN had a negative connection. However, we found no significant correlation between caregiver’s pressure to eat behavior and children’s food choices. This may be due to the unique nature of autistic children, whose food preferences are influenced by many factors. Some researchers have discovered that the symptoms and restrictive behavioral patterns of autistic children can influence their food preferences in addition to other traits like sensory sensitivities and an unwillingness to adapt readily [32]. Additionally, these kids are more likely to be avoidant or restrictive eaters, resulting in good food selectivity, eating monotony, rejection of particular food kinds, and reactions to particular food colors, smells, temperatures, and textures [33]. As well as the symptoms themselves influencing food choices, the food environment associated with eating can also have an impact. There is evidence that the food eaten by caregivers (and the resulting eating patterns) significantly influences the formation of children’s food preferences [34]. The Healthy Eating Index (HEI) and children’s average daily intake of fruits and vegetables were favorably correlated with caregivers’ positive modeling [35,36]. Availability of vegetables and fruits at home may also increase children’s vegetable and fruit intake [37,38]. Although caregivers’ pressure-to-eat behavior does not have a direct effect on the food choices of children with ASD, we do find their force-feeding behaviors significantly increased children’s FN levels, which in turn affected children’s food choices.

The mean score of food neophobia in 4–6 six-year-old preschool children with ASD was 25.56 ± 6.46. In contrast, another study found that among Chinese children with TD, the FN score was 23.73 ± 4.45. Children with autism have a much higher fear of new food than children with TD [39]. Previous studies have found that food neophobia in children generally manifests itself in the form of rejection of healthy foods [40,41], preferring unhealthy, high-energy food [42]. Our study found similar results, with autistic children who were high in food neophobia consuming less fruits, vegetables, and aquatic foods and more snacks. A medical report on a study of 279 children with autism and diagnosed as severely partial eaters showed that at least 67 percent of the children did not eat vegetables, and 27 percent did not eat fruit [43]. This could have to do with the food’s flavor, texture, and aroma. For example, children prefer sour and bitter foods and are naturally drawn to sweet and salty foods [44]. In addition to this, these children are more sensitive to these properties of food due to their sensory sensitivities and, thus, are pickier about food [45]. In addition to the fact that children with ASD are hypersensitive to stimuli that cause them to be reluctant to accept food, children with ASD may have oral motor deficits that cause constipation and diarrhea problems when they swallow food without chewing, which can lead to the refusal of this type of food [46].

Children’s FN is also susceptible to environmental influences. For example, parental food neophobia leads to increased FN in children [47]. In many cases, parents who are having difficulty feeding their children use what they believe to be “strategies” designed to solve existing problems [48]. Specific methods could be incredibly ineffectual and perhaps worsen kids’ maladaptive habits. When parents use food as a reward for an autistic child, it has a negative correlation with the child’s enjoyment of eating, and the child is more likely to refuse food [49]. Our research has found that a more severe caregiver pressure strategy was associated with food neophobia in children with ASD. This is consistent with previous findings [46,50].

Previous research has revealed a favorable association between parental “pressure to eat” feeding techniques and food neophobia in typical preschoolers [39]. The possible causes are as follows: the first is parents’ perception that pressure to eat helps improve children’s diets. According to the results of a previous qualitative study conducted by our research team (yet unpublished), the majority of parents exhibited positive attitudes toward force-feeding. However, the study found that parents who resorted to pressure to eat lowered children’s expectations of the food and rejected it even more [12]. Another reason may be that these children have a higher prevalence of eating behavior problems compared to other children, and their caregivers often believe that the child’s special diet is another feature of the child’s disease. They are more likely to adopt inappropriate feeding behaviors. In future interventions, it is essential to inform caregivers of autistic children that behaviors such as FN and picky eating are not exclusive to these children; TD children also suffer from these eating problems, which may reduce caregivers’ concerns and thus reduce inappropriate feeding behaviors by parents. It could also reduce these children’s food neophobia by informing them that unhealthy feeding practices are not conducive to child growth and development and by making a change in mindset.

Even while most kids with eating disorders get through the first two years of life, kids with ASD are more likely to have eating disorders that are more severe and last a lifetime. In addition to being affected by the disease, high levels of food neophobia in autistic children can affect their dietary intake, and our study found a negative correlation between FN and dietary diversity in children. Poor dietary habits in these children severely limit the diversity of their food intake and the quality of their meals [8] and can result in an increased risk of malnutrition and obesity [51], especially micronutrient deficiencies [52,53]. Food neophobia in ASD children needs more attention than in other children, and more study is required to enhance feeding practices in the future in order to lessen food anxiety and enhance these kids’ nutritional status.

Few studies have looked at the connection between particular feeding practices and autistic children’s FN levels in China. However, this study also has some limitations. First of all, the ASD children investigated in our study were trained by teachers in special institutions in Changsha, which means that the caregivers of the children are better off. They are more concerned about the children’s diet and are willing to participate in this investigation, which may cause some bias. Secondly, the study used a scale questionnaire, which was filled out based on the subjective understanding of the caregiver of the child with ASD. In the future, when investigating this particular group of children, a more objective approach should be taken, such as possibly filling out the questionnaire uniformly by the same classroom teachers after training or developing a relevant tool that can be filled out and assessed by multiple parties. Our study was cross-sectional, and a causal relationship between caregivers’ pressure to eat and FN in these children could not be confirmed. Future studies should include a non-ASD control group to more effectively identify unique or common aspects of feeding behavior and food neophobia in different groups of children.

## 5. Conclusions

We discovered that Chinese children with ASD had higher levels of food neophobia. And the higher the level of social deficits, the higher the food neophobia. The higher the caregivers’ pressure-to-eat score, the higher the level of FN in children, and the more worried caregivers were about their children’s nutritional status, the more likely they were to resort to pressure to eat strategy. Children with ASD who have food neophobia consume less healthy foods such as fruits, vegetables, and aquatic products and more unhealthy foods such as snacks. In addition, food neophobia is positively correlated with children’s dietary diversity.

## Figures and Tables

**Figure 1 children-11-00528-f001:**
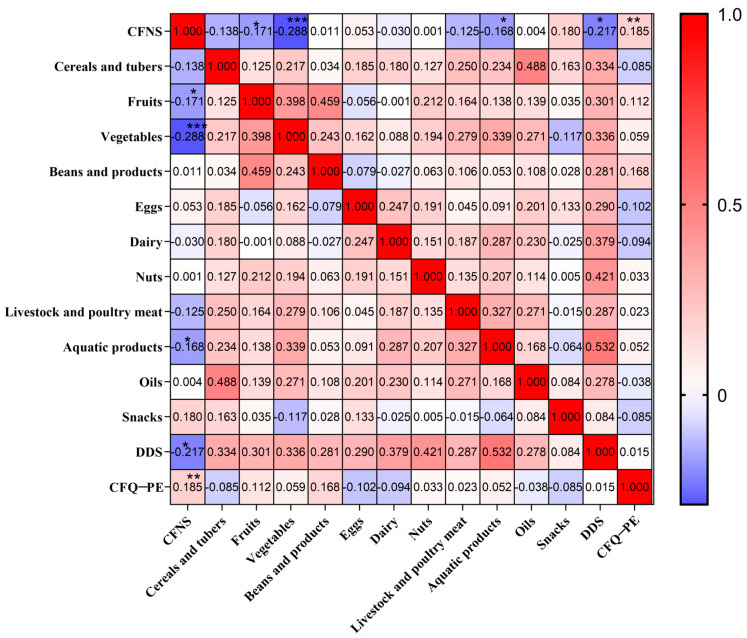
Correlation between caregiver’s pressure to eat, food neophobia, and dietary intake in children with ASD. *—*p* < 0.05, **—<0.01, ***—<0.001. Compared by Pearson correlation analysis.

**Table 1 children-11-00528-t001:** Basic demographics of children with ASD (*n* = 160).

Variables	*n* (%)/Mean (SD)	CFNS Mean (SD)	*p*
SRS score			0.044 *
≤68.00	43 (26.9)	23.91 ± 6.38	
69.00~80.50	37 (23.1)	25.08 ± 6.48	
80.51~98.00	41 (25.6)	25.51 ± 6.05	
≥98.01	39 (24.4)	27.87 ± 6.53	
Age (year)	4.23 ± 1.36		0.806
2~3	52 (32.5)	24.37 ± 6.38	
4~5	82 (51.2)	25.87 ± 6.62	
6~7	26 (16.3)	26.96 ± 5.92	
Sex			0.880
Boys	126 (78.7)	25.52 ± 6.26	
Girls	34 (21.3)	25.71 ± 7.26	
Nationality			0.292
Han	144 (90.0)	25.74 ± 6.53	
Minority	16 (10.0)	23.94 ± 5.70	
Residence			0.561
Urban	141 (88.1)	25.45 ± 6.45	
Rural	19 (11.9)	26.37 ± 6.70	
Only child			0.501
Yes	78 (48.7)	25.91 ± 6.36	
No	82 (51.3)	25.22 ± 6.58	
Circumstances of birth ^#^			0.916
Prematurity	10 (6.3)	26.40 ± 4.25	
Full-term labor	144 (90.6)	25.51 ± 6.23	
Expired property	5 (3.1)	25.60 ± 6.84	
Exclusive breastfeeding duration			0.472
≥6 months	71 (44.4)	25.24 ± 7.12	
<6 months	89 (55.6)	25.96 ± 5.55	
Monthly per capita household income (yuan) ^#^			0.617
<3000	67 (47.9)	25.70 ± 6.12	
3000~5000	43 (30.7)	25.21 ± 8.01	
>5000	30 (21.4)	26.73 ± 6.54	
Educational level of primary caregiver			0.048 *
Middle school and below	48 (30.0)	24.79 ± 6.56	
High school	43 (26.9)	27.63 ± 5.69	
Junior college and above	69 (43.1)	24.80 ± 6.65	
Occupation of primary caregiver			0.148
Employee	44 (27.5)	24.50 ± 6.50	
Leave work and rest	17 (10.6)	26.71 ± 5.32	
Unemployed	69 (43.1)	26.62 ± 6.34	
Other	30 (18.8)	24.00 ± 6.98	

SRS score represents the Social Responsiveness Scale score. *—*p* < 0.05. ^#^—Missing values exist. They were compared by two independent samples, a *t*-test, and one-way ANOVA.

**Table 2 children-11-00528-t002:** General demographics of primary caregivers.

Variables	*n* (%)	PE X ± SD	*p*
Sex			0.727
Male	5 (3.3)	3.25 ± 0.66	
Female	155 (96.7)	3.10 ± 0.96	
Educational level of primary caregiver			0.275
Middle school and below	48 (30.0)	3.17 ± 0.89	
High school	43 (26.9)	3.24 ± 0.93	
Junior college and above	69 (43.1)	2.97 ± 1.00	
Occupation of primary caregiver			0.520
Employee	44 (27.5)	2.97 ± 0.98	
Leave work and rest	17 (10.6)	3.07 ± 0.86	
Unemployed	69 (43.1)	3.23 ± 0.96	
Other	30 (18.8)	3.03 ± 1.05	
Mother’s mode of conception			0.056
Natural conception	156 (97.5)	3.08 ± 0.94	
Test-tube baby	4 (2.5)	4.00 ± 1.02	
Mother’s mode of delivery			0.057
Natural birth (without surgical operation)	82 (51.3)	2.96 ± 1.00	
Cesarean section	78 (48.7)	3.25 ± 0.89	
Exclusive breastfeeding greater than or equal to 6 months of age			0.201
Yes	71 (44.4)	3.02 ± 1.04	
No	89 (55.6)	3.21 ± 0.83	
Fear of child malnutrition			0.006 **
Yes	91 (56.9)	3.28 ± 0.91	
No	69 (43.1)	2.87 ± 0.96	
Particular focus on children’s diets			0.205
Yes	76	3.00 ± 1.00	
No	80	3.19 ± 0.91	
SRS score			0.143
≤68.00	43 (26.9)	2.97 ± 0.90	
69.00~80.50	37 (23.1)	2.93 ± 0.96	
80.51~98.00	41 (25.6)	3.14 ± 1.08	
≥98.01	39 (24.4)	3.38 ± 0.83	

SRS score means the Social Responsiveness Scale score. PE—pressure to eat. It was compared using two independent samples, a *t*-test, and one-way ANOVA. **—*p* < 0.01.

**Table 3 children-11-00528-t003:** Correlation between caregiver feeding behavior and children’s food neophobia.

Variables	*n* (%)/Mean ± SD	Children’s Food Neophobia
Children’s FN	25.56 ± 6.46	-
Restriction (RST)	3.86 ± 0.97	−0.060
Monitoring (MN)	3.82 ± 1.01	−0.032
Pressure to eat (PE)	3.10 ± 0.95	0.185 *
Food as rewards (FR)	3.88 ± 1.21	−0.117
Continue to provide children refused food		0.205 **
Yes	116 (72.5)	
No	44 (27.5)	

*—*p* < 0.05, **—< 0.01. Compared by Pearson correlation analysis.

**Table 4 children-11-00528-t004:** Correlation between parental feeding behavior and children’s food neophobia.

Independent Variables	β	95% CI	*p*
Unadjusted Model			
Constant	20.296	(16.827, 23.766)	<0.001 ***
Pressure to eat	0.193	(0.013, 0.283)	0.013 *
Continue to provide children refused food	0.212	(0.356, 2.128)	0.006 **
Adjusted Model			
Constant	22.706	(16.037, 29.376)	<0.001 ***
Pressure to eat	0.164	(0.078, 2.163)	0.035 *
Continue to provide children refused food	0.211	(0.347, 2.122)	0.007 **

*—*p* < 0.05, **—<0.01, ***—<0.001. We adjusted the following variables in our regression model: restriction, monitoring, food as rewards, educational level of primary caregiver, and SRS score.

## Data Availability

The data presented in this study are available on request from the corresponding author. The data are not publicly available due to privacy.

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
