# Peer review of "The Impact of Caregiver Pressure to Eat on Food Neophobia in Children with Autism Spectrum Disorder: A Cross-Sectional Study"

_children, 2024, doi:10.3390/children11050528_

Round 1
Reviewer 1 Report
Comments and Suggestions for Authors
This is an interesting paper that reports on an important issue. I have some minor and a couple of more substantial comments.
Minor:
As below, the grammar and syntax has some flaws and should be addressed. e.g. lines 50-51 are grammatically incorrect.
Line 42: TD is used without explanation.
More substantial:
A lot of demographic information was gathered from caregivers. The authors should justify each item, . I believe some of these questions (e.g. whether child was breastfed and for how long; whether this is a single parent family) suggest an intention to blame parents.
The authors could reflect on possible ambiguities about who is the primary caregiver. For example, if grandparents influence the behaviour of the parent, who is principally responsible for any dysfunctional behaviours?
Section 2.2.3 reports that a validated questionnaire was adapted for context and expanded with two additional questions before being used in this study. Therefore, the instrument used in the study was not validated, which casts doubt on the validity of the results.
Line 239 "we can also find" is unclear. Does this mean we might find or we do find?
Comments on the Quality of English Language
While the paper is easy to understand, there are some instances of awkward or incorrect grammar that should be addressed via a thorough proof read.
Author Response
|
Dear Reviewer,
We sincerely appreciate your insightful and valuable comments and suggestions. You help us a lot in revising and improving our paper. Here we submit a new version of our manuscript, revised according to the suggestions, titled “The Impact of Caregiver Pressure to Eat on Food Neophobia in Children with Autism Spectrum Disorder: A Cross-Sectional Study” (Manuscript ID: children-2925080). You will see the difference made to the revised manuscript.
If you have any questions about this paper, please do not hesitate to contact me.
Corresponding author: Qian Lin Ph.D. E-mail: linqian @csu.edu.cn Sincerely yours, Qian Lin |

Reviewer 2 Report
Comments and Suggestions for Authors
The manuscript "The Impact of Caregiver Pressure to Eat on Food Neophobia in Children with Autism Spectrum Disorder: A Cross-Sectional Study" investigates the relationship between caregivers' pressure to eat and food neophobia (FN) in children with Autism Spectrum Disorder (ASD). The main finding is that the caregiver’s pressure to eat is associated with a higher level of FN in children with ASD, which negatively impacts their dietary quality. This is an important finding, especially for parents of ASD children.
I have some suggestions that may improve the quality of the paper:
1. Sometimes, especially in tables, it would be good to see abbreviations explained in the legend.
2. I am not sure how regression models were built, how many predictors were in the initial model, only the two presented in the table?
3. The study has no control group. Including a control group of non-ASD children could help determine if the observed effects are unique to children with ASD.
4. The methods are generally clear, but more detailed descriptions of the procedures, e.g. how the food neophobia score is calculated and the specific questions or scales used, would enhance reproducibility and understanding.
Author Response
Dear Reviewer,
We sincerely appreciate your insightful and valuable comments and suggestions. You help us a lot in revising and improving our paper. Here we submit a new version of our manuscript, revised according to the suggestions, titled “The Impact of Caregiver Pressure to Eat on Food Neophobia in Children with Autism Spectrum Disorder: A Cross-Sectional Study” (Manuscript ID: children-2925080). You will see the difference made to the revised manuscript.
If you have any questions about this paper, please do not hesitate to contact me.
Corresponding author: Qian Lin Ph.D. E-mail: linqian @csu.edu.cn
Sincerely yours,
Qian Lin

Reviewer 3 Report
Comments and Suggestions for Authors
Dear authors this is a very interesting study on the association of caregivers’ pressure to eat, food neophobia and dietary intake among children with ASD.
Please clarify this: in the abstract it is mentioned that Information about social demographic characteristics, caregiver’s feeding behavior, autistic children’s food neophobia (FN) scores were collected using an electronic questionnaire. In the Materials and Methods section it is reported that the questionnaire was completed through one-on-one questioning by the investigator. Which is the case?
Furthermore please add to the limitations the cross sectional nature of the study.
Author Response

(The authors gave the same response as above.)

Round 2
Reviewer 2 Report
Comments and Suggestions for Authors
The authors addressed all of my concerns.